# Effects of Mixed Allelochemicals on the Growth of *Microcystis Aeruginosa*, Microcystin Production, Extracellular Polymeric Substances, and Water Quality

**Ping Ouyang [1,2], Chao Wang [1], Peifang Wang [1,*], Xiaorong Gan [1], Xun Wang [1] and Chaohui Yang [1]**

[1] Key Laboratory of comprehensive management of shallow lakes, Ministry of Education, College of environment, Hohai University, No.1 Xikang Road, Nanjing 210098, China; ouyp@jshb.gov.cn (P.O.); cwang@hhu.edu.cn (C.W.); ganxr@hhu.edu.cn (X.G.); xwang2014@hhu.edu.cn (X.W.); 181305020037@hhu.edu.cn (C.Y.)

[2] Climate Change Division, Department of Ecology and Environment of Jiangsu Province, NO. 176 Jiangdong North Road, Nanjing 210036, China

[*] Correspondence: pfwang2005@hhu.edu.cn

**Abstract:** The inhibition of cyanobacteria growth by allelochemicals, which controls harmful algal blooms has been examined in many studies. The objective of this work was to compare the efficiencies of different allelochemicals and determine a mixing proportion corresponding to the highest algae inhibiting activity and smallest adverse effect. The obtained results demonstrated that artemisinin, nonanoic acid, malonic acid, and ethyl acetate inhibited algal growth more efficiently than D-menthol and lactic acid. Synergies were observed in five groups of allelochemical combinations with inhibition ratios exceeding 80%, and the concentrations of extracellular microcystin-LR in the groups with high algal inhibition ratios were lower than that in the control group on the 7th day. No changes in extracellular polymeric substances compositions were detected after treatment. The permanganate indices of the treated groups were higher than that of the control group; however, this disparity gradually decreased with time. In addition, a sharp decrease in the concentration of dissolved inorganic phosphorus was observed for all treated groups. From the obtained data, the optimal proportion of mixed allelochemicals corresponding to 3.94 mg L$^{-1}$ of artemisinin, 6.27 mg L$^{-1}$ of nonanoic acid, 8.2 mg L$^{-1}$ of malonic acid, and 6.38 mg L$^{-1}$ of ethyl acetate was suggested.

**Keywords:** allelochemical mixture; algae inhibition; Microcystin leucine arginine (MC-LR); extracellular polymeric substance (EPS); permanganate index (PI); dissolved inorganic phosphorus (DIP)

## 1. Introduction

In recent decades, the algal blooms caused by the eutrophication in many temperate lakes and reservoirs have been spreading globally [1,2], which represented threat to both ecological and human health [3,4]. Many available methods tried to prevent the occurrence of blooms, including physical, chemical, and biological ways [5]. Among the proposed methods with high efficiency, the use of plant-derived compounds such as allelochemicals has attracted much interest as an environmentally friendly approach [6].

Since the 1990s, the allelopathic properties of aquatic macrophytes and microorganisms have been examined in detail [7]. In particular, isolation and identification of effective substances were performed, and their efficiencies and mechanisms of algae inhibition were investigated. Artemisinin

extracted from *Artemisia annua* produced a strong inhibitory effect on *Microcystis aeruginosa* [8]. In a previous study [9], artemisinin in the form of sustained-release granules inhibited the growth of *Microcystis aeruginosa* with an inhibition ratio (IR) above 95% and production of algal toxins. Nonanoic acid was found to be the most effective inhibitor of the *Microcystis aeruginosa* growth among the fatty acids released from *Myriophyllum spicatum* [10]. Malonic acid produced from yeast extract exhibited strong algicidal properties and inhibited algae growth more effectively when combined with L-lysine [11]. D-menthol possessing antibacterial and antifungal characteristics was found to limit the cellular growth of *Microcystis aeruginosa* during a laboratory test, thus suppressing the increase in the total microcystin (MC) concentration [12]. Lactic acid (2-hydroxypropionic acid) was detected in the culture media of several submerged macrophytes such as *Ceratophyllum demersum* and *Vallisneria spiralis*; it was also used to change the membrane permeability and cell structure of *Microcystis aeruginosa*, decrease its antioxidant capacity, and ultimately cause the crack and death of its cells [13]. In the majority of previous studies, effects of individual inhibitors on algal growth were investigated. However, the synergistic properties of allelochemical combinations and their environmental impacts, such as water qualities, have not been examined in sufficient detail.

The influence of allelochemicals on the production of toxins by algae is a strong determinant of their ecological safety. MCs are the main algal toxins produced by *Microcystis aeruginosa*, which are responsible for the liver failure in wild animals, aquatic species, and humans [14]. MC leucine arginine (MC-LR) is the most toxic and harmful MC of the known algae toxins. Some studies suggested that the application of allelochemicals could promote an increase in the concentrations of extracellular MCs [15]. However, it was also hypothesized that the total MC concentration might decrease due to the continuous stress exerted by these compounds [16,17]. Therefore, the effects of allelochemicals on MC production must be determined to clarify this issue.

Extracellular polymeric substances (EPS) produced via various processes including excretion, secretion, sorption, and cell lysis are mainly composed of polysaccharides, proteins, lipids, and humic substances [18]. They serve as physical barriers between the cells and surrounding environment and play a key role in buffering against the adverse effects of various environmental stressors [19]. At the same time, allelochemicals can partially damage the membrane sheath or cell wall and affect the cell membrane structure, leading to the outflow of intracellular species such as ions, deoxyribonucleic acid, and ribonucleic acid, thus accelerating cell lysis [20]. Therefore, gaining insights into the EPS production and composition is necessary to achieve a better understanding of the influence of allelochemical combinations on *Microcystis aeruginosa*.

In cyanobacteria blooms, algae detritus settles on the surface of the sediment, which increases its carbon, nitrogen, and phosphorus contents [21]. Although various organic compounds and nutrients released by the natural degradation of algae were studied previously [22], the impact of added allelochemicals on contribution of organic substances and bioavailable phosphorus in water have not been investigated in detail. Therefore, this research was carried out to study the inhibiting effects of allelochemicals on *Microcystis aeruginosa*, test the production and release of MCs and EPS and certain water quality indexes during the application of the mixed allelopathy algaecide agent.

## 2. Materials and Methods

### 2.1. Cultivation of Microcystis Aeruginosa

The axenic strain of *Microcystis aeruginosa FACHB-1343* isolated from Taihu Lake in 2010 was provided by the Freshwater Algae Culture Collection of the Institute of Hydrobiology (FACHB) in Wuhan, China. Cultivation was conducted in a sterilized BG-11 medium [23] inside conical flasks at 25 °C under a light/dark regime of 14/10 h. All experimental utensils and medium were autoclaved (MLS-3750, Sanyo, Japan) at 120 °C for 30 min. Three replicates were used for each group. The flasks were shaken twice every day to avoid agglomeration during the experimental period.

### 2.2. Preparation of Allelochemicals and Their Combinations

According to previous studies, concentration gradient experiments were performed for six different chemicals (artemisinin, nonanoic acid, malonic acid, D-menthol, lactic acid, and ethyl acetate) to identify algaecides with high efficiencies in the same batch culture. Ethyl acetate, an organic solvent commonly used for the extraction of allelopathic compounds, was utilized as one of the tested chemicals.

Based on 72-h $EC_{50}$ values and their long-term inhibitory effects on the algae density, four chemicals (artemisinin, nonanoic acid, malonic acid, and ethyl acetate) were selected for orthogonal group testing to study their compound action on the growth of *Microcystis aeruginosa* cells. To determine the optimal proportion of the selected chemicals corresponding to the strongest inhibitory effect, experiments were performed as per the L9 ($3^4$) orthogonal test with the dosing proportions listed in Table 1. Different combinations of the four selected chemicals were added to nine sample groups.

**Table 1.** Dosing proportions of the four selected allelochemicals in the L9 ($3^4$) orthogonal test.

| GroupNo. | Selected Chemicals | | | |
|---|---|---|---|---|
| | Artemisinin | Nonanoic Acid | Malonic Acid | Ethyl Acetate |
| 1 | 1/9 $EC_{50}$ | 1/9 $EC_{50}$ | 1/9 $EC_{50}$ | 1/9 $EC_{50}$ |
| 2 | 1/9 $EC_{50}$ | 1/6 $EC_{50}$ | 1/6 $EC_{50}$ | 1/6 $EC_{50}$ |
| 3 | 1/9 $EC_{50}$ | 1/3 $EC_{50}$ | 1/3 $EC_{50}$ | 1/3 $EC_{50}$ |
| 4 | 1/6 $EC_{50}$ | 1/9 $EC_{50}$ | 1/6 $EC_{50}$ | 1/3 $EC_{50}$ |
| 5 | 1/6 $EC_{50}$ | 1/6 $EC_{50}$ | 1/3 $EC_{50}$ | 1/9 $EC_{50}$ |
| 6 | 1/6 $EC_{50}$ | 1/3 $EC_{50}$ | 1/9 $EC_{50}$ | 1/6 $EC_{50}$ |
| 7 | 1/3 $EC_{50}$ | 1/9 $EC_{50}$ | 1/3 $EC_{50}$ | 1/6 $EC_{50}$ |
| 8 | 1/3 $EC_{50}$ | 1/6 $EC_{50}$ | 1/9 $EC_{50}$ | 1/3 $EC_{50}$ |
| 9 | 1/3 $EC_{50}$ | 1/3 $EC_{50}$ | 1/6 $EC_{50}$ | 1/9 $EC_{50}$ |

*2.3. Algal Inhibition Testing of Allelochemicals and Their Combinations*

Cell density was used to test the inhibition effectiveness of allelochemicals and their combinations. *Microcystis aeruginosa FACHB-1343* was inoculated into 500-mL conical flasks containing 350 mL of the culture medium with an initial algal density range of 5.2–9.5 × $10^6$ cells mL$^{-1}$. In the concentration gradient experiments, artemisinin, nonanoic acid, malonic acid, D-menthol, lactic acid, and ethyl acetate were added separately to the flasks yielding concentrations of 0, 5, 10, 20, 40, 60, 80, and 120 mg L$^{-1}$. The algal cell density was determined under a light microscope (Zeiss Axioskop 40, Shanghai) using a hemocytometer (Dark-line, Marienfeld). Samples were taken on the first, third, fifth, eights, twelfth, and seventeenth days. IR was calculated via the following formula:

$$IR(\%) = 100(C_1 - C_2)/C_1 \tag{1}$$

where $C_l$ (cells mL$^{-1}$) is the cell density in the control group, and $C_2$ (cells mL$^{-1}$) is the cell density in the treated group.

During group orthogonal testing, proportions of various added components were determined from the 50% effective concentrations measured after 72 h (72-h $EC_{50}$) for four selected chemicals (identified by concentration gradient experiments as artemisinin, nonanoic acid, malonic acid, and ethyl acetate). Their specific proportions used in nine treatments are listed in Table 1, where 1/9, 1/6, and 1/3 represent different fractions of the 72-h EC50 values. Samples were taken and tested on 3rd, 5th, 7th days.

*2.4. Measurements of Extracellular MC-LR Concentrations*

Twenty milliliters of algal suspension were centrifuged at a speed of 6000 r min$^{-1}$ and temperature of 4 °C for 10 min, after which the supernatant was filtered through a 0.22-mm GF/C glass fiber membrane. The filtrate was dried under nitrogen at 40 °C and adjusted to a constant volume of 1 mL using high-performance liquid chromatography (HPLC)-grade methanol. An HPLC

system (e2695, Waters) was utilized to measure the MC-LR content. The wavelength was set to 238 nm, the injection volume was 50 μL, and the flow rate was 0.6 mL min$^{-1}$. The mobile phase consisted of 0.1% formic acid (wt.%) in water (solvent A, 40%) and 0.1% formic acid (wt.%) in HPLC-grade methanol (solvent B, 60%). MC-LR standard was purchased from Agent Technology (Switzerland), and the established standard curve ($R^2$ > 99.9%) covered a concentration range from 1 to 1000 μg L$^{-1}$.

### 2.5. Extraction and Characterization of EPS

Extraction of EPS was performed according to the method reported by Xu and Jiang [24]. A total of 20 mL *Microcystis aeruginosa* culture was centrifuged at 6000 r min$^{-1}$ for 15 min, and the harvested algal cells were dissolved in a 0.05% NaCl (wt.%) solution and centrifuged at 12000 r min$^{-1}$ for 15 min. The supernatant was collected for EPS measurements. Fluorescence analysis was conducted by a fluorescence spectrometer (Hitachi F-7000, Hitachi High Technologies, Tokyo, Japan) in the scan mode at 25 °C using a 700-voltage xenon lamp. Emission (Em) spectra were recorded from 250 to 650 nm by varying the excitation (Ex) wavelength from 200 to 500 nm. The scan rate was 1200 nm min$^{-1}$, and the wavelength intervals of Em and Ex were 5 nm each. The instrument response and inner-filter effect corrections were performed to adjust for instrument specific biases. The excitation-emission matrix (EEM) spectra of Milli-Q water were subtracted from the EEM spectra of the EPS samples.

### 2.6. Water Quality Assays

Permanganate index (I$_{Mn}$), a synthetic indicator widely used in the environmental monitoring system in China, represents a degree of the organic pollution of surface water. The allelochemicals added in our experiment are all organic substances. Using them as algal inhibitors would inevitably lead to the increase of organic matter content in water. Thus, the extent of the increase in organic pollution is of great concern to our study. In many research papers, total nitrogen and total phosphorus are usually discussed to indicate the degree of water eutrophication. For most shallow lakes in China (especially the Taihu Lake where experimental strain of *Microcystis aeruginosa FACHB-1343* isolated from), the phenomenon of phosphorus excess is more than that of nitrogen excess. Not all forms of phosphorus can be directly utilized by cyanobacteria, while dissolved inorganic phosphorus (DIP), as the most important source of phosphorus that can be directly absorbed by plants and algae, impacts on the proliferation of cyanobacteria and algal bloom control. Therefore, I$_{Mn}$ and DIP were used in our study as the water quality indicators.

Potassium permanganate is used to oxidize organic and inorganic reductive substances in water under certain conditions, and the equivalent amount of oxygen is calculated from the amount of potassium permanganate consumed [25]. The analysis method based on I$_{Mn}$ represents the national standard of People's Republic of China (GB 11892-89). In this work, 15 mL of the algae suspension was taken and diluted to 100 mL in a conical flask followed by the sequential addition of sulfuric acid (1+3) and potassium permanganate solution (0.01 mol L$^{-1}$). The resulting mixture was heated in water to 98 °C for 30 min, after which 10 mL of sodium oxalate solution (0.01 mol L$^{-1}$) was added to it. Potassium permanganate solution (0.01 mol L$^{-1}$) was dosed into the flask as part of the standard titration procedure until the mixture turned pink without color fading in the next 30 s. The consumed volume of the standard titration solution was recorded as $v_1$. Subsequently, 100 mL of ultrapure water was used as a control sample to repeat this step twice, and the consumed volumes of the standard titration solution were recorded as $v_0$ and $v_2$. I$_{Mn}$ was calculated via the following formula:

$$I_{Mn} = -8 + 16(15 + 10v_1 - 8.5v_0)/v_2 \tag{2}$$

DIP sample was obtained by filtrating the algae suspension through 0.45-mm PTFE membranes (Xinya Purification Materials Co., Shanghai, China). Concentrations of DIP were assayed by a molybdenum blue spectrophotometric method using ultraviolet radiation [26].

### 2.7. Statistical Analysis

Statistical analysis was performed using IBM SPSS 19.0 software for Mac OS X. The data obtained in this study were presented as means ± standard deviations (SD, n = 3). EC$_{50}$ value of each

chemical tested in concentration gradient experiments were calculated using probit regression. Significant differences of algae cell densities, IR ratios, concentrations of MC-LR, DIP and $I_{Mn}$ idexes among different treatment groups were analyzed using one-way ANOVA followed by LSD post hoc test at level of $p < 0.05$.

## 3. Results

*3.1. Inhibitory Effects of Six Different Chemicals on the Cell Density of Microcystis Aeruginosa*

The changes in the algae cell density observed during each treatment are shown in Figure 1. The cell numbers in groups exposed to artemisinin gradually decreased throughout the experiment (Figure 1a), except for the treatments at concentrations of 5 and 10 mg L$^{-1}$. In these two groups, slow and sharp increases in the *Microcystis aeruginosa* cell densities were observed during the first 8 d and in the latter period, respectively. Compared with the control group, the cell growths in the groups treated with 5, 10, 20, and 40 mg L$^{-1}$ of artemisinin were inhibited in a dose-dependent manner. No significant differences ($p = 0.018$ on the 12th day, $p = 0.6$ on the 17th day) in the cell density were observed among the groups treated at artemisinin concentrations of 60, 80, and 120 mg L$^{-1}$ after 12 days of exposure.

In Figure 1b,c,f, algae inhibition was observed at the initial stage in the presence of nonanoic acid, malonic acid, and ethyl acetate, as the cell densities in those treatment groups at low dose (5 mg L$^{-1}$) decreased to 66%, 93% and 80% of that in the control group, respectively. However, the inhibition effect weakened with a rapid increase in the cell density after 8 d of exposure at concentrations of 5, 10, and 20 mg L$^{-1}$ and ultimately reached a level ($\geq$56.05 mg L$^{-1}$ in nonanoic acid treated group, $\geq$35.74 mg L$^{-1}$ in malonic acid treated group, $\geq$46.21 mg L$^{-1}$ in ethyl acetate treated group) close to or even higher than that of the control group. The cell densities in the groups treated with nonanoic acid, malonic acid, and ethyl acetate at concentrations greater than 60 mg L$^{-1}$ remained below initial density during the entire experiment, suggesting that effective long-term algal inhibition was achieved through high-dose treatments.

The *Microcystis aeruginosa* growth was slightly stimulated with algae cell densities approaching (8.65 mg L$^{-1}$ on the 3rd day) or even suppressing (9.65 mg L$^{-1}$ on the 3rd day) that of the control (8.64 mg L$^{-1}$ on the 3rd day) during treatments at concentrations 10 mg L$^{-1}$ and 20 mg L$^{-1}$ in the D-menthol exposure groups, respectively (Figure 1d). Algae inhibition effects were observed on the first day of treatment among the groups with high dose, as cell densities in groups with 80 mg L$^{-1}$ and 120 mg L$^{-1}$ D-menthol decreased to 73% and 69% of that in the control group, respectively. The cell growth began to recover and became vigorous after approximately 5 days of culture, indicating that the allelopathic inhibitory effect weakened in the long term.

In Figure 1e, the data of algae cell densities in groups treated with low and moderate doses ($\leq$60 mg L$^{-1}$) of lactic acid (14.38–16.48 × 10$^6$ cells mL$^{-1}$ on the 3rd day) were very close to that in the control group (17.53 × 10$^6$ cells mL$^{-1}$ on the 3rd day). High dosages corresponding to the concentrations equal to or exceeding 80 mg L$^{-1}$ apparently inhibited the algae growth during the first 12 days, after which the effect decreased dramatically with the consumption of lactic acid after cultivation. On the 3rd day, the cell densities in treatment groups (lactic acid concentration $\geq$ 80 mg L$^{-1}$) were only 29%–32% of that in the control group. On the 17th day, this percentage data raised to 51%–78%.

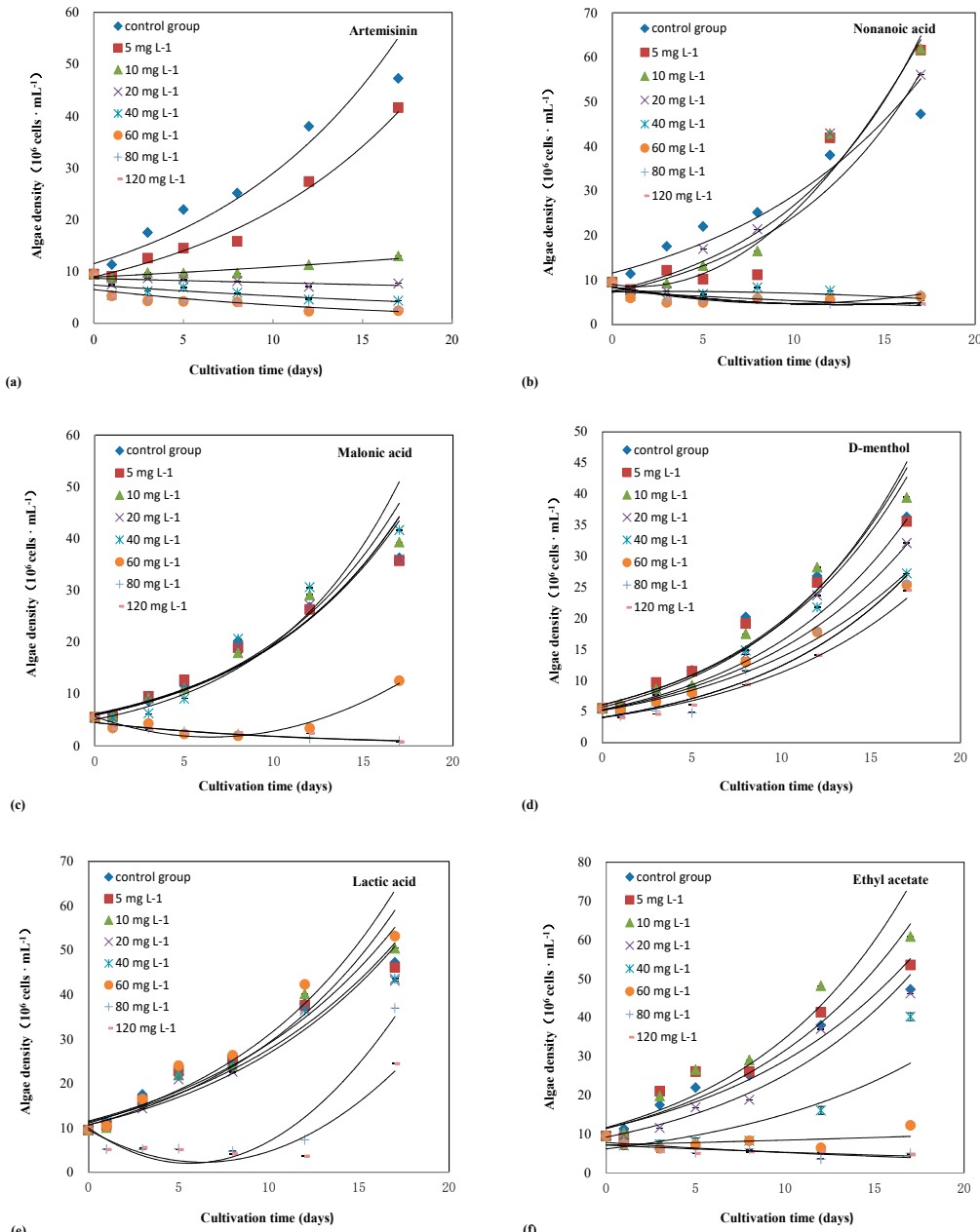

**Figure 1.** Cell density of *Microcystis aeruginosa* in different treatments. Figure 1(**a**)–(**f**) presented algal inhibition effects in groups treated by artemisinin, nonanoic acid, malonic acid, D-menthol, lactic acid and ethyl acetate, respectively.

The statistic results of algae cell densities ($10^6$ cells mL$^{-1}$) in concentration gradient experiments on the 3rd day were shown in Table 2. The 72-h EC$_{50}$ values of the six chemicals were calculated using regression equations (Table 3). They were equal to 23.66 (artemisinin), 18.81 (nonanoic acid), 73.8 (malonic acid), 115.72 (D-menthol), 86.21 (lactic acid), and 38.26 (ethyl acetate) mg L$^{-1}$.

**Table 2.** The statistic results of algae cell densities ($10^6$ cells mL$^{-1}$) in concentration gradient experiments on the 3rd day (n = 3).

| Group | Artemisinin | | | Nonanoic Acid | | | Malonic Acid | | |
|---|---|---|---|---|---|---|---|---|---|
| | $\overline{X} \pm S$ | F | P | $\overline{X} \pm S$ | F | P | $\overline{X} \pm S$ | F | P |
| 0 mg L$^{-1}$ | 17.53 ± 0.09 | | | 17.53 ± 0.09 | | | 8.64 ± 0.19 | | |
| 5 mg L$^{-1}$ | 12.56 ± 0.37 | | | 12.13 ± 0.88 | | | 9.57 ± 0.37 | | |
| 10 mg L$^{-1}$ | 9.77 ± 0.22 | | | 9.24 ± 0.72 | | | 9.25 ± 0.72 | | |
| 20 mg L$^{-1}$ | 8.71 ± 0.41 | 1291.949 | P < 0.001 | 7.41 ± 0.41 | 203.552 | P < 0.001 | 9.16 ± 0.84 | 120.959 | P < 0.001 |
| 40 mg L$^{-1}$ | 6.17 ± 0.06 | | | 5.57 ± 0.51 | | | 6.27 ± 0.14 | | |
| 60 mg L$^{-1}$ | 4.32 ± 0.09 | | | 4.91 ± 0.44 | | | 4.33 ± 0.09 | | |
| 80 mg L$^{-1}$ | 4.34 ± 0.15 | | | 5.11 ± 0.58 | | | 3.35 ± 0.15 | | |
| 120 mg L$^{-1}$ | 4.87 ± 0.10 | | | 5.81 ± 0.20 | | | 3.19 ± 0.29 | | |
| **Group** | **D-menthol** | | | **Lactic Acid** | | | **Ethyl Acetate** | | |
| | $\overline{X} \pm S$ | F | P | $\overline{X} \pm S$ | F | P | $\overline{X} \pm S$ | F | P |
| 0 mg L$^{-1}$ | 8.64 ± 0.19 | | | 17.53 ± 0.09 | | | 17.53 ± 0.09 | | |
| 5 mg L$^{-1}$ | 9.65 ± 0.37 | | | 16.21 ± 0.23 | | | 21.03 ± 1.14 | | |
| 10 mg L$^{-1}$ | 8.65 ± 0.27 | | | 15.70 ± 0.23 | | | 19.79 ± 0.23 | | |
| 20 mg L$^{-1}$ | 7.70 ± 0.72 | 120.959 | P < 0.001 | 14.38 ± 0.45 | 319.506 | P < 0.001 | 11.67 ± 0.69 | 393.855 | P < 0.001 |
| 40 mg L$^{-1}$ | 6.60 ± 0.32 | | | 15.80 ± 0.69 | | | 7.38 ± 0.72 | | |
| 60 mg L$^{-1}$ | 6.59 ± 0.58 | | | 16.48 ± 0.29 | | | 6.35 ± 0.10 | | |
| 80 mg L$^{-1}$ | 5.08 ± 0.53 | | | 5.18 ± 0.85 | | | 6.26 ± 0.22 | | |
| 120 mg L$^{-1}$ | 4.54 ± 0.44 | | | 5.58 ± 0.53 | | | 5.42 ± 0.46 | | |

**Table 3.** EC$_{50}$ (mg L$^{-1}$) and regression equation of six chemicals against *Microcystis aeruginosa*.

| Chemicals | Regression Equation Probit (Y) = b × logarithm (X) + a | 95% Confidence Limits | EC$_{50}$ |
|---|---|---|---|
| Artemisinin | Y = 0.016X − 0.384 | b 0.016~0.016 a −0.385~−0.383 | 23.66 |
| Nonanoic acid | Y = 0.029X − 0.537 | b 0.029~0.029 a −0.539~−0.535 | 18.81 |
| Malonic acid | Y = 0.017X − 1.262 | b 0.017~0.017 a −1.265~−1.258 | 73.8 |
| D-menthol | Y = 0.018X − 2.044 | b 0.018~0.018 a −2.047~−2.041 | 115.72 |
| lactic acid | Y = 0.034X − 2.943 | b 0.034~0.034 a −2.944~−2.942 | 86.21 |
| Ethyl acetate | Y = 0.013X − 0.49 | b 0.013~0.013 a −0.492~−0.488 | 38.26 |

### 3.2. Inhibitory Effects of Mixed Allelochemicals on the Growth of Microcystis aeruginosa Cells

The inhibition ratio (IR) values obtained for each group on third, fifth, and seventh days are presented in Figure 2. It shows that the IRs of all groups increased gradually. On the 3rd day, groups 4, 6, and 9 exhibited stronger inhibition effects than those of the other groups with IR values exceeding 50%, indicating that the observed synergies were caused by combining the selected allelochemicals at the dosing proportions based on their 72-h EC$_{50}$ values. On the seventh day of the experiment, the IRs of groups 1, 6, and 9 maintained their high values exceeding 80%. Therefore, it can be concluded that groups 6 and 9 were treated with the optimal combinations of allelochemicals corresponding to the strongest anti-algal effect. The allelochemical concentrations were 3.94 (artemisinin), 6.27 (nonanoic acid), 8.2 (malonic acid), 6.38 (ethyl acetate) mg L$^{-1}$ in group 6 and 7.89 (artemisinin), 6.27 (nonanoic acid), 12.3 (malonic acid), and 4.25 (ethyl acetate) mg L$^{-1}$ in group 9, respectively.

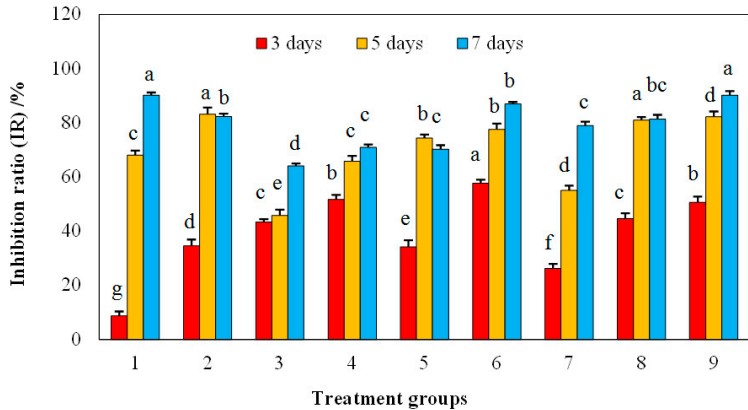

**Figure 2.** Influences of nine different treatments on the IR of *Microcystis. aeruginosa* during orthogonal testing. On the 3rd, 5th, and 7th days, significant difference of IRs among treatment groups was marked with letters a–g, respectively (p < 0.05).

### 3.3. Influence of Mixed Allelochemicals on Extracellular MC-LR Content

The extracellular MC-LR concentrations measured on the 3rd, 5th, and 7th days of the experiment are shown in Figure 3. Their values determined for all samples (including the MC-LR content in the control group) increased consistently during the entire experimental period. On the 3rd day, the control group reached a peak of 14.58 µg L$^{-1}$, suggesting that the dosing of allelochemical combinations reduced the secretion of MCs in the treated groups by decreasing the cell densities at the initial stage. Afterwards, the contents of extracellular MC-LR in the majority of the treated groups increased much more rapidly than those in the control group, which indicated that larger amounts of MC-LR were released from allelochemicals into the surrounding water under the external pressure. Until the 7th day, the MC-LR contents in groups 1, 3, 4, 5, and 7 peaked at 59.86, 84.99, 48.38, 83.22, and 43.59 µg L$^{-1}$, respectively, which were much higher than the value of 24.35 µg L$^{-1}$ obtained for the control group. At the same time, the MC-LR concentrations in groups 6, 8, and 9 were 28.59, 23.37, and 19.11 µg L$^{-1}$, close to or even lower than that of the control group. The allelochemical concentrations were 3.94 (artemisinin), 6.27 (nonanoic acid), 8.2 (malonic acid), 6.38 (ethyl acetate) mg L$^{-1}$ in group 6; 7.89 (artemisinin), 3.14 (nonanoic acid), 8.2 (malonic acid), 12.75 (ethyl acetate) mg L$^{-1}$ in group 8; and 7.89 (artemisinin), 6.27 (nonanoic acid), 12.3 (malonic acid), 4.25 (ethyl acetate) mg L$^{-1}$ in group 9, respectively.

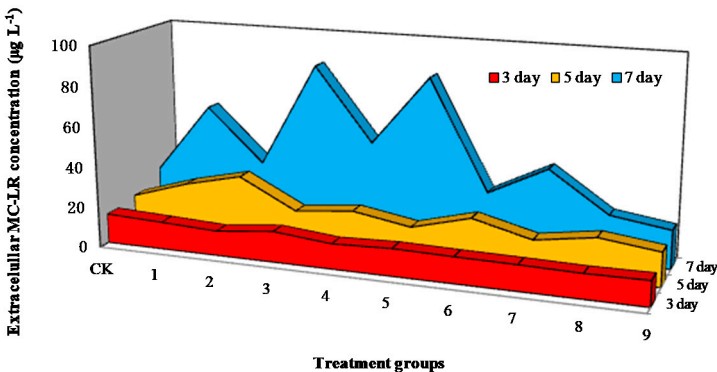

**Figure 3.** Influence of mixed allelochemicals on the extracellular microcystin leucine arginine (MC-LR) concentration in each sample group.

### 3.4. Influence of Mixed Allelochemicals on EPS Characterization

The three-dimensional EEM fluorescence spectra of the loosely bound EPS produced by *Microcystis aeruginosa* in each group were recorded on the 3rd day. The typical EEM contours and detailed fluorescence intensities are presented in Figure 4 and Table 4, respectively. Three fluorescence peaks were consistently detected for the EPS in each group. The first peak corresponded to Ex/Em = 255/285 (peak A), which was attributed to the presence of tyrosine and protein-like compounds. The second peak (peak B) was in the range of Ex/Em = 280 − 285/315 − 335 and represented the fluorescence signals of pure tryptophan and tryptophan containing protein-like compounds. The third peak was located at Ex/Em = 250/390 (peak C), corresponding to fulvic acid-like substances. Compared with the control, variations of the intensities of peaks B and C were observed. More specifically, stronger fluorescence intensities of tryptophan and fulvic acid-like substances were detected for the treated groups. Among the treated groups, slight variations in the peak intensities were observed, but no apparent differences in the peak positions of EPS were detected.

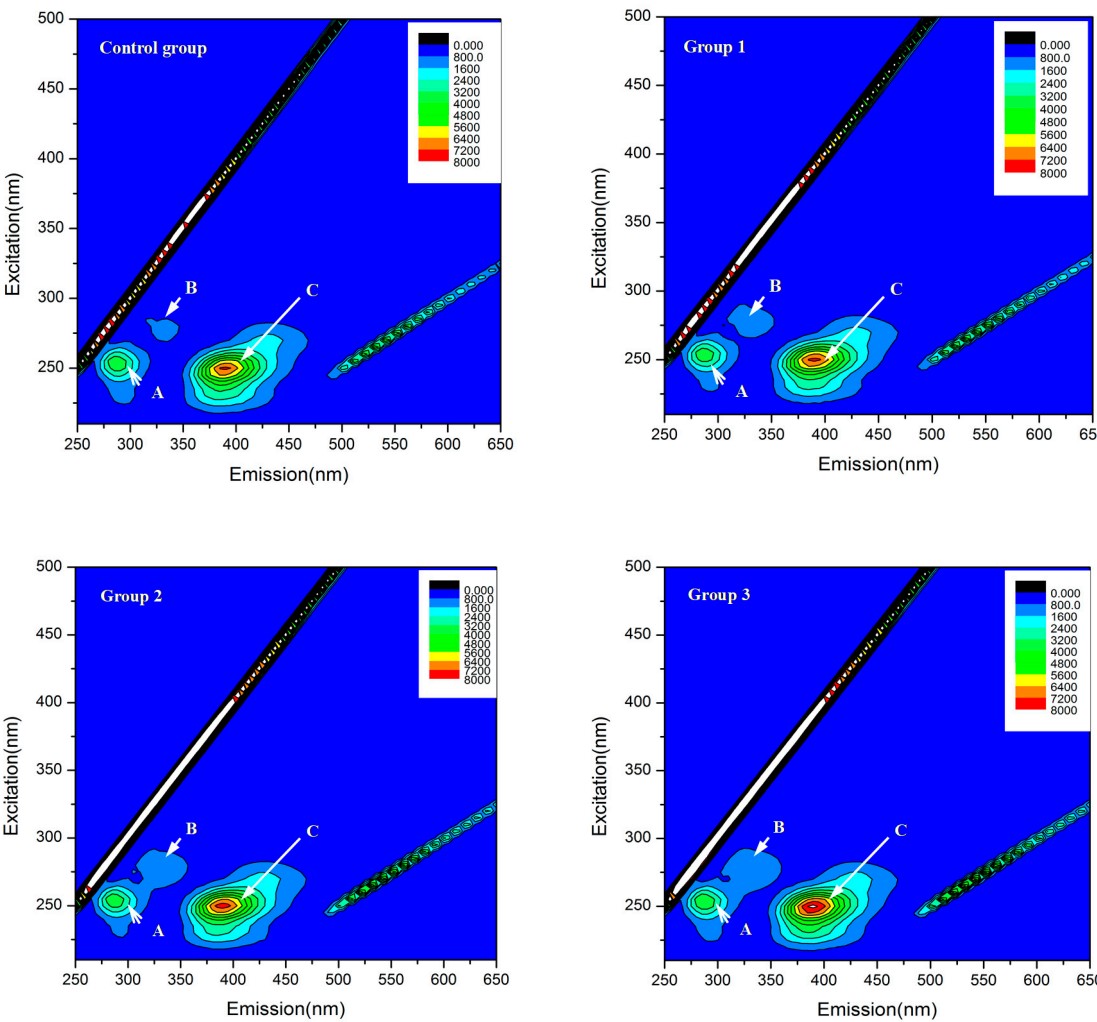

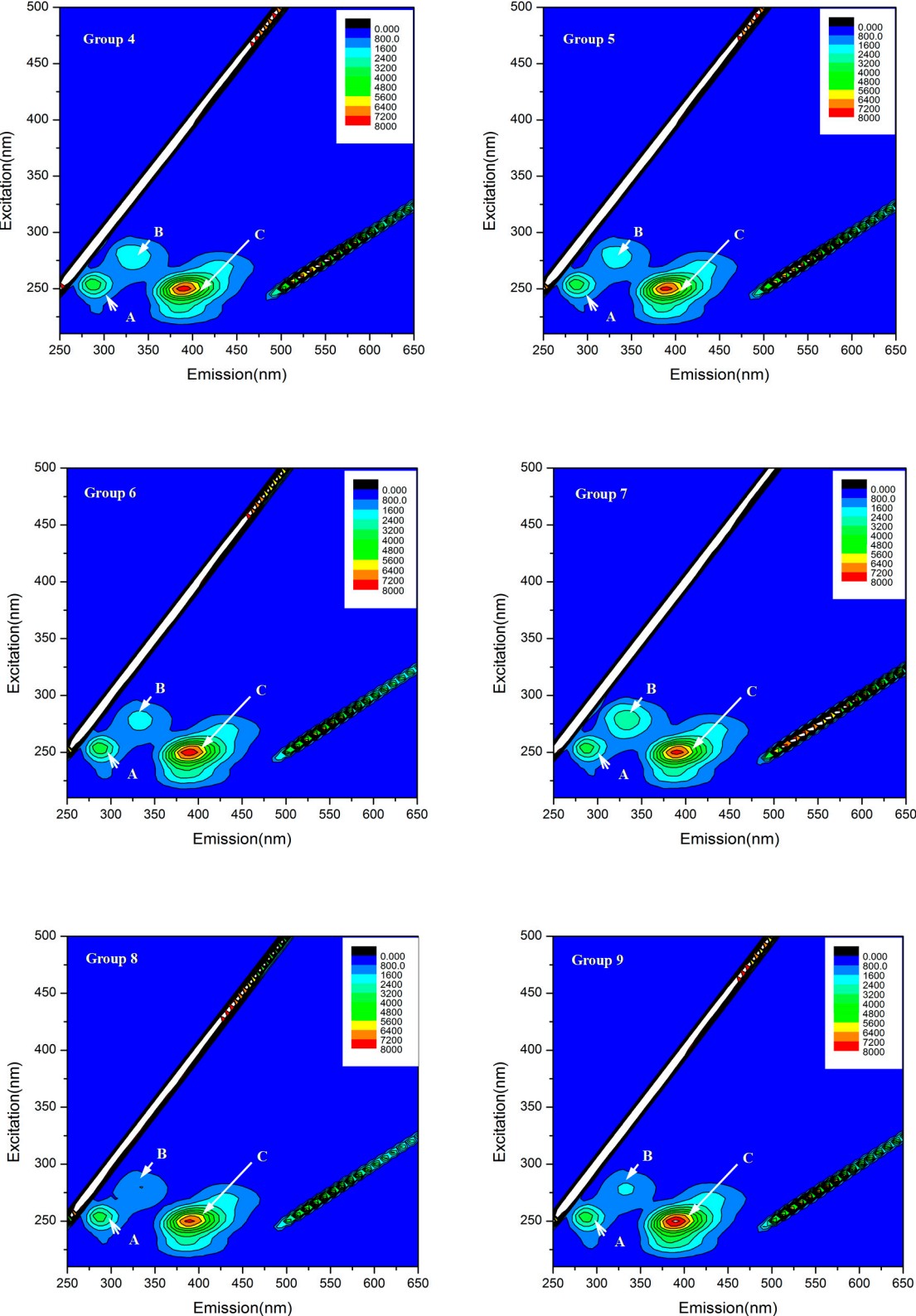

**Figure 4.** Typical excitation-emission matrix (EEM) contours of the various extracellular polymeric substance (EPS) fractions in control group and group 1–9. Peaks A, B, and C were obtained at Ex/Em combinations of 255/285, 280–285/315–335, and 250/390, respectively.

**Table 4.** Fluorescence spectral parameters of the EPS after 3 days of exposure to mixed allelochemicals.

| Group No. | Peaks | Ex/Em | Intensity | Corresponding Peaks |
|---|---|---|---|---|
| Control | A | 255/285 | 3890 | Tyrosine and protein-like compounds |
| | B | 285/315 | 822 | Tryptophan containing protein-like compounds |
| | C | 250/390 | 7397 | Fulvic acid-like compounds |
| 1 | A | 255/285 | 3825 | Tyrosine and protein-like compounds |
| | B | 280/335 | 1089 | Tryptophan |
| | C | 250/390 | 7489 | Fulvic acid-like compounds |
| 2 | A | 255/285 | 3888 | Tyrosine and protein-like compounds |
| | B | 280/335 | 1229 | Tryptophan |
| | C | 250/390 | 7694 | Fulvic acid-like compounds |
| 3 | A | 255/285 | 3963 | Tyrosine and protein-like compounds |
| | B | 285/320 | 1149 | Tryptophan containing protein-like compounds |
| | C | 250/390 | 8291 | Fulvic acid-like compounds |
| 4 | A | 255/285 | 3835 | Tyrosine and protein-like compounds |
| | B | 280/335 | 2226 | Tryptophan |
| | C | 250/390 | 7765 | Fulvic acid-like compounds |
| 5 | A | 255/285 | 3832 | Tyrosine and protein-like compounds |
| | B | 280/330 | 2506 | Tryptophan |
| | C | 250/390 | 8002 | Fulvic acid-like compounds |
| 6 | A | 255/285 | 3866 | Tyrosine and protein-like compounds |
| | B | 280/335 | 2971 | Tryptophan |
| | C | 250/390 | 7798 | Fulvic acid-like compounds |
| 7 | A | 255/285 | 3715 | Tyrosine and protein-like compounds |
| | B | 280/335 | 3004 | Tryptophan |
| | C | 250/390 | 7855 | Fulvic acid-like compounds |
| 8 | A | 255/285 | 3876 | Tyrosine and protein-like compounds |
| | B | 280/335 | 1837 | Tryptophan |
| | C | 250/390 | 8339 | Fulvic acid-like compounds |
| 9 | A | 255/285 | 3869 | Tyrosine and protein-like compounds |
| | B | 280/335 | 2015 | Tryptophan |
| | C | 250/390 | 8465 | Fulvic acid-like compounds |

*3.5. Influence of Mixed Allelochemicals on $I_{mn}$ and DIP Concentrations*

The effects of mixed allelochemicals on DIP and $I_{Mn}$ are illustrated in Figure 5. According to Figure 5a, the concentrations of DIP in the treated groups increased rapidly during the first three days, ranging from 5.77 to 8.58 mg L$^{-1}$ on the 3rd day, then decreased at longer exposures. On the 7th day, DIP concentrations in nine treated groups were significantly lower ($p < 0.05$) than that in the control group (2.31 mg L$^{-1}$). Compared with groups 1, 4, 7, 9 and control group, the concentrations of DIP in groups 2, 3, 5, 6 and 8 on the 7th day were relatively lower, ranging from 0.41 to 0.53 mg L$^{-1}$.

In this experiment, the four selected chemicals are carbon-containing organic compounds. As shown in Figure 5b, the $I_{Mn}$ values of all treated groups (ranging from 57.59 to 61.51, from 53.70 to 56.69, from 58.95 to 62.67 mg L$^{-1}$ on the 3rd, 5th, and 7th day, respectively) were higher than that of the control group (36.53, 42.50, 55.89 mg L$^{-1}$ on the 3rd, 5th, and 7th day, respectively) throughout the entire experiment, indicating that the addition of mixed allelochemicals increased the chemical oxygen demand of the algae suspension. Meanwhile, the $I_{Mn}$ of the control group increased continuously, which likely resulted from the photosynthesis and proliferation of cyanobacteria. By the 7th day, the $I_{Mn}$ values of the control group and groups 3, 5, and 8 were 55.89, 58.95, 60.57, and 59.5 mg L$^{-1}$, respectively, which were significantly lower than the magnitudes obtained for the other

treated groups (p < 0.05). However, this disparity narrowed with time, indicating that the adverse effect on the organic contents would be almost acceptable in the long run.

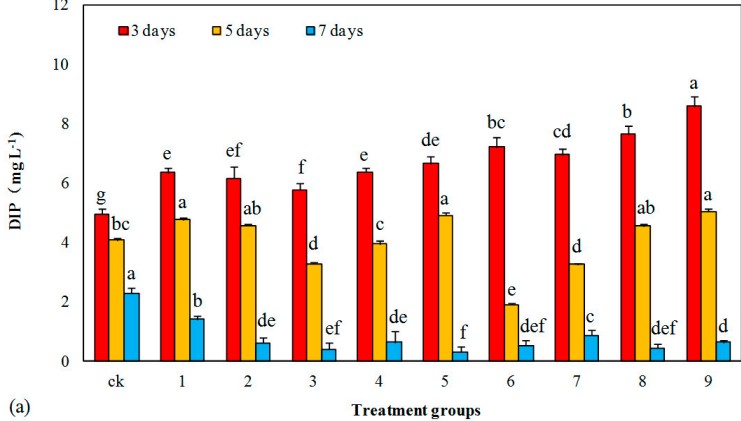

(a)

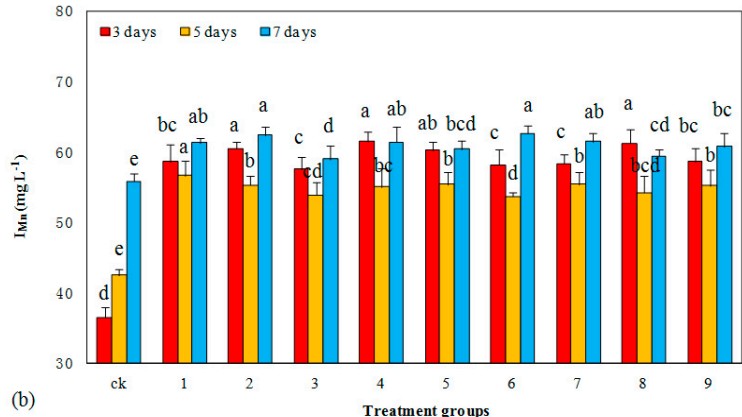

(b)

**Figure 5.** Influences of different allelochemical combinations on the water quality during orthogonal testing. (**a**) Dissolved inorganic phosphorus (DIP) concentration in the supernatant and (**b**) $I_{Mn}$ of the suspension. On the 3rd, 5th, and 7th days, significant difference of DIP and $I_{Mn}$ among treatment groups was marked with letter a–g, respectively (p < 0.05).

## 4. Discussion

### 4.1. Contribution of the Allelochemicals to the Allelopathic Effect of Microcystis Aeruginosa

It is concluded that growth of *Microcystis aeruginosa* was promoted in artemisinin exposure groups at low dose, which was in good agreement with the studies of Ni et al. [27]. Explosive growth of algal cells after approximately 8 days of culture showed that the algal cell growth began to recover due to either the degradation or transformation of the effective anti-algal components [28]. In groups under exposure of nonanoic acid, malonic acid, and ethyl acetate, algae proliferation was observed to be enhanced by the low doses of certain allelochemicals and inhibited by the high doses. Moreover, the inhibitory effect on the algal growth did not change significantly with a dose increase once the concentration range exceeded a 60 mg L$^{-1}$ in groups mentioned above, indicating the existence of a threshold effect [29]. Interestingly, ethyl acetate, an extracting agent commonly used to extract allelochemical compounds from aquatic plants [30], was proven to apparently inhibit the growth of *Microcystis aeruginosa* cells at larger dosages exceeding 40 mg L$^{-1}$ in our study, suggesting that the additional role of ethyl acetate as an algaecide should be taken into account during allelochemical extraction.

In our study, the effect of algal inhibition resulted from D-menthol and lactic acid was relatively poor compared with the other four selected allelochemicals. However, Hu et al. [12] reported that D-menthol effectively inhibited the growth of *Microcystis aeruginosa* cells at a dosage of 3333 mg $L^{-1}$. The wide gap between the dosing concentrations is likely the main reason for the different results obtained in both studies. Peng et al. [13] reported that the IR of 100 $\mu L^{-1}$ lactic acid for the *Microcystis aeruginosa* growth reached 90.6% after 72 hours at an initial cell density of $1.3 \times 10^6$ cells $mL^{-1}$. However, the dose-response relationship was observed only at higher concentrations in our study, which might be due to the different initial cell density.

## 4.2. Compound Effects on Microcystis Aeruginosa Growth and Extracellular MC-LR

Compound effects caused by the coexistence of allelochemicals are often more complex than those of single substances and may be manifested as antagonistic, additive, or synergistic ones [31]. In our study, inhibition ratio in groups 4, 6, and 9 exceeded 50% on the 3rd day, presenting relatively higher efficiency than those in single allelochemical treatments, as the dosage was designed according to 72-h $EC_{50}$ values of each selected ones. Similarly, Ni et al. [32] reported a synergistic effect for catechol and pyrogallic acid mixed at equipotent concentrations.

It was reported previously that the application of chemical algae-killing reagents could deactivate the algal cells or cause their death and disruption, leading to the release of a substantial amount of intracellular algal toxins [33]. The observed increase in the extracellular MC-LR content in our study might be attributed to two factors. First, allelochemicals could promote the release of intracellular microcystins (MCs) by the living *Microcystis aeruginosa* cells into the medium. Second, the addition of allelochemicals could lyse the cells, further promoting the release of intracellular MCs [12]. For groups 6, 8, and 9, higher IR magnitudes were obtained during the entire experimental period. Low algae densities in these groups reduced the secretion of MCs by cyanobacteria keeping their MC-LR contents at low levels, indicating that the long-term effective inhibition of algae growth was related to algal toxin control, which was in good agreement with the findings of Liu et al. [20].

## 4.3. Possibility of EPS Variations of Microcystis Aeruginosa

EPS are composed of large quantities of aromatic compounds and unsaturated fatty chains with fluorescence characteristics [34]. In our study, peaks in treated groups existed at the same range as the control, while concentrations of substances at peak B and C, which represented tryptophan containing protein-like compounds and fulvic acid-like substances, was noted to be enhanced. Meanwhile, increasing trend of extracellular MC-LR concentration in treated groups was observed as we discussed above. Gan had reported that EPS production and the expression of some polysaccharide synthesis genes in microcystis could be promoted by increasing concentration of MCs in environment [35]. The secretion of certain EPS compounds by *Microcystis aeruginosa* was expected to be more sensitive during the algae inhibition process than that in the normal culture course. The increase in the EPS content might be related to the cell lysis [36] caused by cyanobacteria under toxic conditions as a protective response to toxicants [18].

Thus, it can be speculated that dosing allelochemical combinations in the *Microcystis aeruginosa* medium would make little influence on EPS composition but promotes production. Therefore, from the security risk viewpoint, the addition of mixed allelochemicals would not introduce new types of fluorescent compounds (especially extracellular polysaccharide) to the cyanobacterial blooming water after algae inhibition and detritus decomposition.

## 4.4. Impacts on Water Quality

The accumulation of dead cyanobacteria caused by the blue-green algae bloom makes the sediments rich in active organic matter [37] and turns the water body into a pool of endogenous nitrogen and phosphorus species [38]. Techer et al. [6] performed a mesocosm study on the allelopathic potentials and ecotoxicity of gallic and nonanoic acids, in which substantial increases in ammonia and orthophosphate concentrations were observed. The above-mentioned works mainly

examined the release and transformation of carbon and phosphorus species during the degradation of cyanobacteria; however, our study investigated the content variations of the organic matter and DIP directly used by the aquatic life.

As reported by Shapiro [39], cyanobacteria are widely distributed prokaryotic photoautotrophic organisms, which use chlorophyll as the main photosynthetic pigment to consume $CO_2$ through the Calvin cycle. In our experiment, the continuous proliferation of algae in the control group narrowed the organic content gap between the treated and control groups as culture time went on. Phosphorus affects the physiology of cells through a range of cellular processes including protein, sugar, and nucleic acid metabolism [40]. In our experiment, the slight increase of DIP at the early stage and sharp decrease in the later period was likely caused by the leakage of phosphorus from decomposable active substrates and its adsorption onto cell residuals as reported by Li [41]. A similar conclusion was drawn by Sun who reported that the colloids and particulates produced by cyanobacteria decomposition might limit the increase in the concentration of dissolved ionic nutrients due to biosorption [42].

## 5. Conclusions

In this study, we tested cell densities of *Microcystis aeruginosa* when exposed to six different chemicals. It was proved that there were big differences in algal inhibition efficiency among the six inhibitors. The order of anti-algal efficiency from high to low was: nonanoic acid, artemisinin, ethyl acetate, malonic acid, lactic acid, D-menthol. The first four chemicals were selected and mixed in nine different proportions in our orthogonal test. Analyses of inhibition ratios confirmed that the combined action of allelochemicals would result to different algal inhibition effects depending on the species, quantity and mixture ratio. Mixture in groups 6 and 9 got extra attention corresponding to the strongest anti-algal effect during the whole experiment period. By testing microcystin-LR in all groups, concentrations of MC-LR in all treatment groups were found to increase gradually as time went by. However, the concentrations of MC-LR in groups 6, 8, and 9 were observed to be similar to that in the control group on the 7th day, which was resulted comprehensively by promotion and inhibition of microcystin release through cell lysis and biomass control under the stress of mixed allelochemicals. EPS analysis confirmed that there was no difference in the types of algal inhibition products among groups treated by different proportions of mixed allelochemicals. In view of the composition of allelochemicals and impact on bioavailable phosphorus, water qualities represented by indicators of $I_{Mn}$ and DIP were tested in our study. It was confirmed that the addition of allelochemical combinations would decrease the DIP concentration in water, but slightly increase $I_{Mn}$ at longer exposures. In groups 3, 6, and 8, DIP concentrations and $I_{Mn}$ values were relatively lower than those in the other groups. Taking the high algal inhibiting efficiency and environmental friendliness into account, optimal dosage proportions in group 6 was identified: 3.94 (artemisinin), 6.27 (nonanoic acid), 8.2 (malonic acid), 6.38 (ethyl acetate) mg $L^{-1}$. These results can help to elucidate the mechanism of algal inhibition and feasibility of applying mixed allelochemicals to naturally eutrophicated water in future studies.

**Author Contributions:** Conceptualization, P.O. and C.W.; methodology, P.O. and P.W.; software, P.O.; validation, X.W., C.Y., and X.G.; formal analysis, P.O.; investigation, X.G.; resources, P.W.; data curation, C.Y.; writing—original draft preparation, P.O.; writing—review and editing, X.W.; visualization, P.O.; supervision, P.W.; project administration, C.W.; funding acquisition, C.W. All authors have read and agreed to the published version of the manuscript.

**Funding:** This research was funded by the Key Program of the National Natural Science Foundation of China (grant No. 41430751), the National Science Funds for Creative Research Groups of China (grant No. 51421006), the National Major Projects of Water Pollution Control and Management Technology (grant No. 2017ZX07204003), and the National Natural Science Foundation of China (grant No. 51579073).

**Acknowledgments:** Technical support provided by Changzhou ecological environment monitoring center was acknowledged.

**Conflicts of Interest:** The authors declare no conflicts of interest.

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
