# Peer review of "Effects of Mixed Allelochemicals on the Growth of Microcystis aeruginosa, Microcystin Production, Extracellular Polymeric Substances, and Water Quality"

_water, doi:10.3390/w12071861_

Round 1

Reviewer 1 Report

On line 34 is - Wanget al. 2020, it should be [5]

on line 54, the autors write: However, the synergistic properties of allelochemical combinations and thier environmental impacts have been examined in sufficient detal. Studing the environmental effects od allelochemical combinations used is very important.

It is a pity that the autors do nat provide basic water parameters such sa: pH, dissolved oxygen content, etc. as these parameters changed diuring the test.

Figures: 2; 3; 5 are not clear, please change the color or texture.

Reviewer 2 Report

See attached file.

Round 2

Reviewer 2 Report

There is still some red coloured text in lines 189-190.